# Evaluating the safety and patient impacts of an artificial intelligence command centre in acute hospital care: a mixed-methods protocol

Ciarán McInerney [1,2] Carolyn McCrorie,[2,3] Jonathan Benn,[2,3] Ibrahim Habli,[4] Tom Lawton,[5] Teumzghi F Mebrahtu [1] Rebecca Randell [2,6] Naeem Sheikh,[2] Owen Johnson[1,2]

## ABSTRACT

**Introduction** This paper presents a mixed-methods study protocol that will be used to evaluate a recent implementation of a real-time, centralised hospital command centre in the UK. The command centre represents a complex intervention within a complex adaptive system. It could support better operational decision-making and facilitate identification and mitigation of threats to patient safety. There is, however, limited research on the impact of such complex health information technology on patient safety, reliability and operational efficiency of healthcare delivery and this study aims to help address that gap.

**Methods and analysis** We will conduct a longitudinal mixed-method evaluation that will be informed by public-and-patient involvement and engagement. Interviews and ethnographic observations will inform iterations with quantitative analysis that will sensitise further qualitative work. Quantitative work will take an iterative approach to identify relevant outcome measures from both the literature and pragmatically from datasets of routinely collected electronic health records.

**Ethics and dissemination** This protocol has been approved by the University of Leeds Engineering and Physical Sciences Research Ethics Committee (#MEEC 20-016) and the National Health Service Health Research Authority (IRAS No.: 285933). Our results will be communicated through peer-reviewed publications in international journals and conferences. We will provide ongoing feedback as part of our engagement work with local trust stakeholders.

For numbered affiliations see end of article.

**Correspondence to**
Dr Ciarán McInerney;
C.McInerney@leeds.ac.uk

**Strengths and limitations of this study**

► The study design was informed by patient and public representatives.
► The study will use rich electronic health records data and real-time information to assess patient safety and flow.
► This is a longitudinal mixed-method evaluation study designed to focus on both outcomes and organisational behaviour, informed by a multidisciplinary coinvestigator team.
► The study may be limited in its capability to distinguish between contributions motivated by the response to COVID-19 and the impact of command centre.

nuclear power plants and air traffic control systems, nevertheless demonstrate how safety can be maintained.[7 8]

Such high-reliability organisations use centralised 'mission control' structures to integrate intelligence in real time and facilitate safe and efficient operational decision making. Within acute care settings, a hospital 'command centre' or 'mission control' is a relatively novel concept but one that is now achievable given recent advances in electronic health records, data integration and investment from commercial developers. There is, however, limited research on the impact of such complex health information technology on patient safety, reliability and operational efficiency of healthcare delivery. It is also unknown whether such centralised structures are appropriate for coordinating a hospital given that (1) healthcare provision is not bounded within the hospital, unlike energy provision within a nuclear power plant, (2) the system under 'control' is composed of people, unlike non-living materials in off-shore oil and (3) healthcare

## INTRODUCTION

Fragmented healthcare delivery adversely affects patient safety and care.[1 2] Health information technology might help information flow across fragmented organisations[3 4] but investment and adoption has been limited, and is not a completely sufficient solution.[5] The result is a complex sociotechnical organisation that challenges healthcare delivery.[6] In the face of dynamic risks and organisational complexity, high-reliability organisations, like

operates on a demand-pull dynamic, rather than the supply-push dynamic.

The sociotechnical requirements for effective command centres have been studied in transport and military situations,[9] and there have been attempts to use artificial intelligence (AI) principles within command centres since Project Cybersyn in Chile as early as 1971.[10] There is a limited evidence base for this form of digital technology in healthcare[11]—although some successes have been reported in the USA[12]—but there is an increasing belief, supported by the UK Government's Life Sciences Strategy, that AI should play a key role in transforming and modernising the National Health Service (NHS) in the UK.[13 14] Thus, there is a need to understand how hospitals manage their operations through advances in health information technology, and how this affects the quality and safety of patient care.

There is a mismatch between the factors required for successful implementation, and end users and provider perspectives.[15] The disruptions to workflow are significant challenges for users, particularly in systems that have limited modularity and configurability.[15] There have been very few published attempts to evaluate effectiveness, except for use as a clinical decision support tool[16] and most of the evidence originates from North America.[17] Given the major differences in the social, political and economic foundations of their healthcare system, it is important to explore whether these issues are relevant to the UK context.[18]

Even less is known about the potential of electronic health records developments to improve patient outcomes. Patient journeys are poorly understood, with aggregate statistics obfuscating detail. What has not been achieved is real-time command and control using the data generated by routine systems, despite a rise in the number of state-of-the-art dashboards, flow and simulation models.

Bradford Teaching Hospitals NHS Foundation Trust provides hospital services for around half a million people in the district of Bradford, West Yorkshire, UK. In 2019, the Trust implemented a hospital command centre, working with commercial suppliers from the USA. Located at the Bradford Royal Infirmary hospital, this command centre is made up of GE Kryptonite software and display screens (also known as 'tiles'). The tiles serve various functions according to the needs of the organisation and its patients—some display or summarise information pulled directly from hospital computer systems, while others provide augmented intelligence by interpreting data, such as in the 'deteriorating patient' tile that highlights patients at high risk of poor outcomes. Predictive models and automated algorithms generate metrics for display on the tiles in order to provide new insights to support human decision making, based on pre-existing data, including information that hospital staff enter into electronic health records.

The Bradford AI command centre aims to provide faster and safer care by reducing unnecessary waiting by anticipating and avoiding bottlenecks in care delivery before they cause problems. Such AI command centres have the potential to improve future patient flow and safety, and research to understand the health service delivery, safety and operational factors should be considered an area of major importance for hospitals.

We hypothesise that the implementation and integration of a real-time, centralised hospital command centre will improve patient flow, enhance situational awareness, support operational decision making and facilitate identification and timely mitigation of threats to patient safety. Supported by funding from the National Institute for Health Research (NIHR), we have proposed a mixed-methods project that combines ethnographic observations, qualitative process evaluation and quasi-experimental methods to study the evolving sociotechnical nature of the systems and processes within hospitals. Our protocol is theoretically informed by contemporary safety science theory concerning system resilience,[19 20] human factors models of situational awareness[21 22] and command-and-control in high reliability organisations.[23–25]

### Aim and objectives

Our aim is to evaluate the safety and patient impacts of the Bradford AI command centre. Our objectives are:

1. To evaluate the impact of the command centre on patient safety, hospital operational efficiency and related organisational processes ('Impact Evaluation').
2. To understand the process of implementation and integration of the command centre with data infrastructure and organisational processes ('Process Evaluation').
3. To contextualise the findings through cross-sector and cross-industry perspectives on hospital command and control technologies ('Cross-industry perspectives').
4. To synthesise findings into practical outputs to engage service stakeholders and inform future investment and practice ('Synthesis and dissemination').

Further breakdown of the objectives is shown in online supplemental appendix 1, indicating the contribution of the research activities described in the next section.

## METHODS AND ANALYSIS
### Design and data collection

System implementations such the Bradford AI command centre are complex interventions into complex adaptive systems that could provide improvements but might also result in emergent unforeseen consequences.[26 27] We will conduct a longitudinal mixed-method evaluation informed by the multidisciplinary co-investigator team and public and patient involvement and engagement. A mixed-method approach is well suited to study complexity interventions[28] and the complex adaptive systems to which they are applied. Mixed-method approaches have been used to study information flow and organisational networks,[29] integration of organisational interventions,[30 31] effectiveness of service models,[32] and how

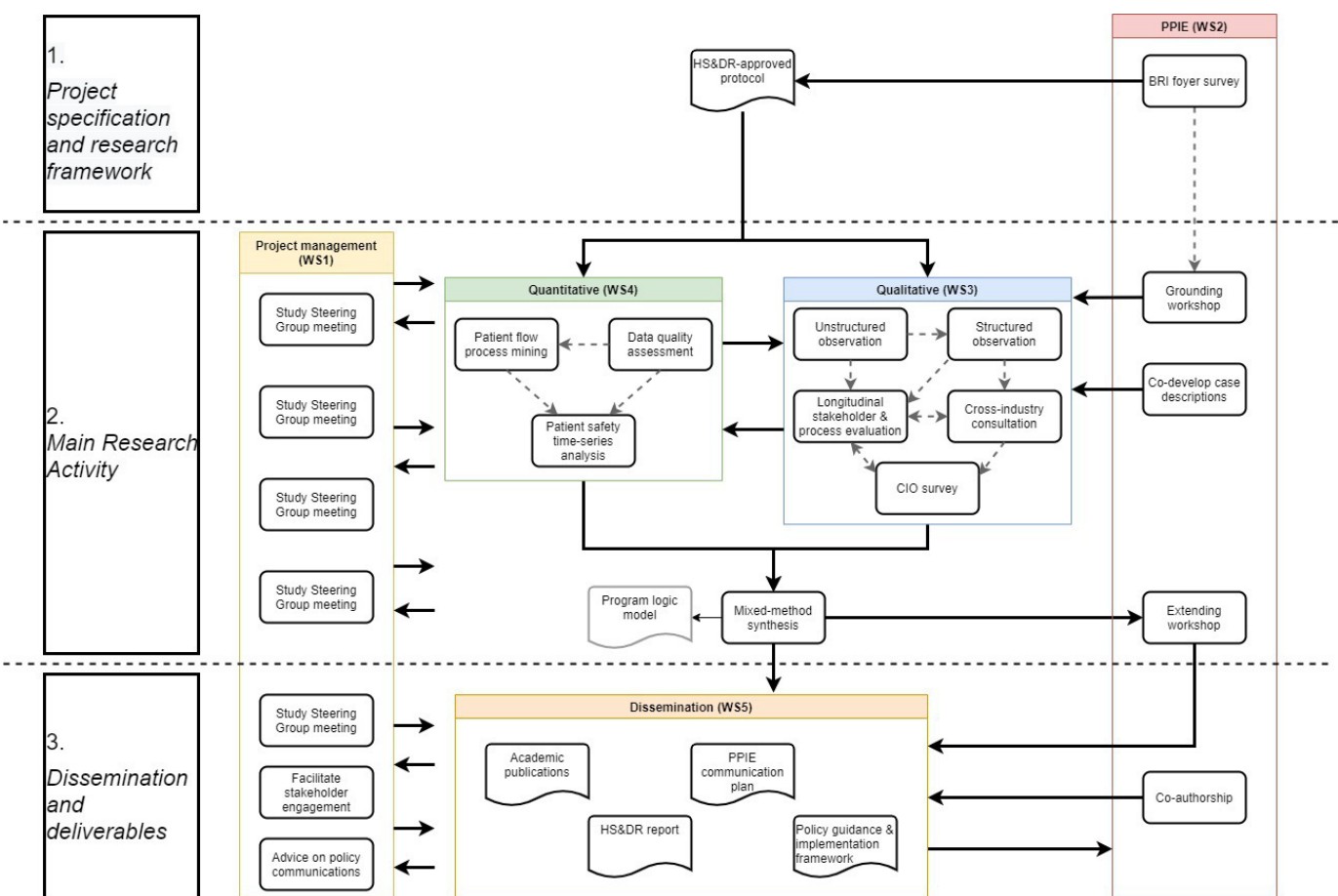

**Figure 1** Schematic overview of the research project's components. BRI, Bradford Royal Infirmary; CIO, chief information officer; HS&DR, health service and deliver research funding programme; PPIE, patient and public involvement and engagement; WS, work stream.

health information technology affects communication,[33] patient monitoring,[34–36] care provision[37] and clinical decision making.[38]

Our study comprises five substudies across five work streams (figure 1). The five substudies are conducted by the qualitative and quantitative work streams (WS3 and WS4 in figure 1). These work streams mutually inform each other as part of an iterative synthesis of findings, rather than solely a summative synthesis. Qualitative and quantitative work streams will work in parallel, with qualitative interviews and ethnographic observations informing iterations for quantitative analysis that will sensitise further qualitative work. Quantitative work will take an iterative approach to identify relevant outcome measures from the literature and pragmatically from the dataset. These will be subsequently verified by the qualitative work.

The main research activity will be guided by the project's Study Steering Group (see WS1 in figure 1) and by patient-and-public involvement and engagement (PPIE; see WS2 in figure 1). Our study steering group is an independent body ensuring the project is conducted to the rigorous standards set out in the Department of Health's Research Governance Framework for Health and Social Care and the Guidelines for Good Clinical Practice.[39]

The Study Steering Group will also facilitate stakeholder engagement in research and dissemination, and will advise on policy communications (see Ethics and dissemination, below). Membership of our Study Steering Group includes clinical, technical, commercial and academic healthcare representatives from the UK, Canada, USA, China, Australia (online supplemental appendix 2).

### Substudy 1: data quality

This substudy will contribute to objectives 1 ('Impact Evaluation') and 2 ('Process Evaluation') based on the hypothesis that the introduction of the command centre will affect the awareness of, recording of and processing of electronic healthcare data, which are inextricably related to data infrastructure, operational efficiency and organisational processes. We will apply Weiskopf et al's 3×3 matrix to assess the quality of healthcare data.[40] This framework maps Patient, Variables and Time data items in terms of Completeness, Correctness and Currency (in other words, presence, accuracy, and timeliness). Further detail on how to implement the 3×3 matrix is available in Weiskopf et al.[41] Our initial identification of variables will be informed by our qualitative substudy (substudy 4); case-description work will define the command centre tiles of interest. The data presented or used to inform

variables that are presented on these tiles will then be requested as data abstracts. We will require clinical input to determine the expected attributes of the variables of interest. For example, if a patient has a weekly timestamp in their record but blood pressure is only expected to be taken fortnightly, then we will not consider empty entries every other week as incomplete.

## Substudy 2: patient flow

This substudy will contribute to objectives 1 ('Impact Evaluation') and 2 ('Process Evaluation') based on the hypothesis that operational efficiency, organisational processes and patient safety are affected by the flow of patients through the hospital. To study patient flow, we will use process-mining methods[42] to describe patients' journeys through their hospital care.[43] We will construct process models to represent patients' logs of clinical events (see examples in dentistry,[44] oncology,[45] sepsis,[46] and primary care[47]). We will evaluate these models by comparing their performance when constructed using various process-mining algorithms. The performance of the models will be measured by[48]:

1. Replay fitness: a measure of how many traces from the log can be reproduced in the process model, with penalties for skips and insertions; range 0–1.
2. Precision: a measure of how 'lean' the model is at representing traces from the log. Lower values indicate superfluous structure in the model; range 0–1.
3. Generalisation: a measure of generalisability as indicated by the redundancy of nodes in the model. The more redundant the nodes, the more variety of possible traces that can be represented; range 0–1.

Patient flow as defined by the best-performing process model will be described using the multi-level approach of Kurniati et al,[49] which include activity, trace and model measures. As all substudies progress, other measures of patient flow might be suggested for study, for example, patient-level measures.

## Substudy 3: patient safety

This substudy will contribute to objectives 1 ('Impact Evaluation') and 2 ('Process Evaluation') by directly evaluating the differences in patient-safety outcomes before and after the implementation of the command centre. The evaluation will use longitudinal data-analysis methods, for example, including interrupted time-series analysis and latent growth modelling. We will model trends behaviour before, during and after the implementation of the command centre, with consideration for the onset of the COVID-19 pandemic. Unobserved confounders will be handled by including a control site from the same geographical region that uses the same electronic health record system, but which does not use a command centre.

We will approach the analysis in a responsive manner, adding or removing interrupts in response to unfolding understanding of the command centre's implementation, based on our qualitative process evaluation. Candidate

| Table 1 | List of proposed variables for analysis | | |
| --- | --- | --- | --- |
| Variable | Patient flow | Patient safety | Data quality |
| Ambulance diversion rates | x | | |
| Ambulance handover times | x | | |
| Cancelled operations (electives and non-electives) | x | | |
| Completeness | | | x |
| Correctness | | | x |
| COVID bed availability | x | | |
| Currency | | | x |
| Diagnostic process time | x | | |
| Early discharges | x | | |
| Falls in hospital | | x | |
| Hospital-acquired infections | | x | |
| In-hospital transfers | x | | |
| Intensive care unit bed usage | x | | |
| Left without being seen rates | | x | |
| Length of stay | x | | |
| Marked 'hospital discharge' | | | x |
| Mortality in hospital | | x | |
| Mortuary crowding | x | | |
| No of patients awaiting surgery (inpatients/at home) | x | | |
| Postoperative sepsis rate | | x | |
| Pressure sores in hospital | | x | |
| Readmission rates for same condition (within 48–72 hours) | | x | |
| Time to admission | x | | |
| Time to be seen | x | | |
| Time to discharge | x | | |
| Time to treat stroke patients | | x | |
| Waiting time benchmarks, for example, 4 hours/18 hours | x | | |

variables of interest will include the Patient Safety Indicators from the Agency for Healthcare and Research Quality, for example, pressure ulcer rate, in-hospital fall with hip fracture rate, post-operative sepsis rate.[50] These will be supplemented by variables of interest informed by the qualitative substudies and early PPIE workshops. See table 1 for the potential list of variables for analysis. The final included set of variables will be defined based on availability of historic data, data quality and relationship with the intervention logic, as established through the parallel qualitative work.

We assume that patient safety is influenced by the flow of patients and the quality of information (as encoded in electronic health records). Therefore, we also intend to use the aforementioned substudies on data quality and patient flow to inform clinically meaningful outcomes

logically related to patient safety. These outcomes will be subject to longitudinal analysis like the patient safety outcomes.

## Substudy 4: ethnography and qualitative interview

This study will contribute to objectives 1 ('Impact Evaluation') and 2 ('Process Evaluation'). Ethnographic enquiry has been selected to facilitate deep understanding of the technology in its broader social and organisational context, including human experience, engagement and interaction.[51 52] We aim to achieve a comprehensive description of how the command centre is integrated and embedded within the broader sociotechnical hospital system through observation of enacted working practices, communication, decision making and operating culture. In this sense, we will not simply be relying on the model for the system implementation as planned by programme leads, but will explore the differences between work as intended and work as done[19] and describe any unintended consequences and implementation barriers as they emerge. The ethnographic and qualitative interview substudy is composed of three main research activities, detailed below.

### Unstructured observations

We will conduct unstructured observations in the command centre to gather information of staff interactions with data, and the ways in which this influences decision making. Through research field notes and interviews, we will seek to understand the role of the command centre in coordinating care, from the perspective of staff in and around the command centre. Two researchers will undertake up to 36 hours of observation, completed in up to 4-hour windows that represent different times of day and days of the week. Observation periods will be prespecified through arrangement with command centre leads. In addition, the researchers will record incidents of observer effects to allow analysis of whether participants' awareness of the researchers' presence changed over time.[53] We will also explore behaviour and meaning around specific events, drawing on the critical incident technique.[54]

We will review emerging hospital policies and guidance related to the command centre (eg, meeting minutes and operational procedures) where practicable as an alternative to data collection involving staff. A sampling framework to guide collection of documents will be informed through earlier qualitative interviews with command centre leads and iterated during the research process. Documents that meet inclusion criteria will be recorded in a document inventory and a data extraction template will be created to obtain the necessary information from the documents. The extracted data will be analysed through an inductive process to capture key developments in design and functioning of the command centre.

### Longitudinal stakeholder and process evaluation

Our process evaluation framework[55] will systematically explore the experiences, beliefs and expectations from multiple user perspectives in relation to operational planning and delineate the trajectories by which patient safety, operational and other intermediary outcomes are impacted by command centre processes.

We will use qualitative research interviews at multiple time points within the command centre programme. This will include building on the ethnographic work to explore interactions with prior theory concerning how the command centre might work. The evaluation will address the factors that govern engagement with and use of this technology, using technology adoption theory.[56] The evaluation will also address the efficiency and effectiveness of processes for generating new intelligence for decision making and quality improvement, at the level of the hospital. A key output will be a logic model to describe the command centre as a complex, health-informatics intervention.

Sampling will be theoretically driven, based on emerging insights from the structured observations, and will include command centre programme leads, key roles working in the centre, clinical leads in frontline areas interacting with the command centre and organisational-level stakeholders representing senior information systems, operational strategy, clinical governance and financial interests. Up to 20 interviews will be undertaken at the study site focusing on two timepoints: one during the early phase of the project and the second towards the end of data collection. Representation of comparable roles will be sought at the control site, for comparative analysis of how the implicated functions are delivered in conventional operational planning processes.

### Structured observations

We will undertake structured observations to inform use-cases of operational planning, control and decision making in priority areas, at the hospital level. Our approach to structured observation will draw on engineering use-case methodology[57] to understand usability of the system in context. We will follow key information from the command centre's visual displays to decision and actions taken by key professional roles. This will involve shadowing various roles, such as bed managers, risk management, quality assurance, clinical leads and others, as they act and make decisions based on Command Centre information. We will produce up to six use cases, based on 60 hours of observation. We will use the use cases identified in structured observations as a probe to compare operational planning, control and decision making in specific priority areas, with and without the support of a centralised command centre function, to enrich our understanding of how a command centre operates within a health service context.

Field notes and interview transcripts will be entered into NVivo (V.12.6) software to facilitate data management. Data analysis will comprise both inductive and

deductive analyses, employing frameworks derived from prior theory and a comparative perspective across the two study sites.[58 59]

Relevant theoretical frameworks for sense making in our analysis will include models of situational awareness,[60] operational command and control,[61–63] sociotechnical evaluation,[64] high reliability organisations[7 62 65] and resilience in healthcare.[66 67] From the perspective of situational awareness theory, for example, we will seek to understand how the command centre enhances human perception of the environment and events within time and space, including projection of future states, and facilitates comprehension of meaning.[60]

### Substudy 5: cross-industry consultation and chief information officer survey

The substudy will contribute to objective 3 ('cross-industry perspectives'). We will undertake a cross-industry review as part of our work, comprising qualitative literature review and consultation with subject-matter experts in a range of safety-critical domains. Such an approach has been applied successfully in previous work, which sought to elicit and apply knowledge from high-risk industry to the development of incident reporting systems in healthcare.[68]

Data collection will involve scoping the literature in a range of domains for conceptual and empirical models of causal mechanisms for centralised command and control. We will additionally consult with up to 10 industry experts, including representatives of similar command centre programmes in other health systems, accessed through UK healthcare human factors and other professional networks. This process will be supported by the project's Study Steering Group. The results will be synthesised to inform analysis and interpretation of our data.

We will also conduct a survey of the perceptions of NHS information personnel in acute care across England and Wales to understand variations in electronic-data-facilitated command and control within hospitals beyond the two research sites. The survey instrument will be informed by earlier qualitative research work and literature review. We will capture views on current practices in data-supported operational planning and control, the costs-benefits of investment in centralised command and control 'centres', information/data readiness, implementation barriers and perceptions of the need for further development in this area.

The survey will be broadcast to NHS information personnel through the University of Leeds Online Surveys network. Data will be transferred into SPSS (v24) software to facilitate data analysis.

### Sampling and recruitment

For the qualitative work, we will recruit relevant staff at the command centre and control site in key roles. NHS staff working in and around the command centre will be asked to take part in ethnographic observations. Up to 40 NHS staff will be interviewed, 20 from each site. We will sample ≤10 cross-industry experts for interviews, and ≤40 NHS hospital information personnel in England and Wales will be asked to take part in the survey. Sampling will be theoretically informed in accordance with qualitative research practices to maximise variation in stakeholder perspectives. Potential participants will be identified through clinical leads and early observations.

For the quantitative work stream, we will use complete sampling of electronic health records within relevant periods. The duration of relevant periods will be informed by the initial case description and unstructured observations in the qualitative work, which will sensitise us to the information handled by the command centre.

### Consent and data handling

Research participants will be told that they do not have to take part in the research if they wish and that they can withdraw up to the point that their data has been anonymised (<2 weeks following research interviews; <1 week following survey). The quantitative work will analyse routinely collected healthcare records data. These data will have been deidentified and processed by the hospitals' data teams and accessed via Connected Yorkshire—an ethically approved regional integration of healthcare and other data available for research purposes.[69]

### Patient and public involvement and engagement (PPIE)

PPIE is an integral part of our study design, delivery and dissemination. Figure 1 shows how the PPIE work stream is engaged throughout all phases of the project. Pre-study PPIE included a patient and public representative as co-applicant (NS), input on research design by the PPIE Research Fellow at the NIHR Yorkshire and Humber Patient Safety Translational Research Centre, and an informal survey of visitors to the hospital in which the command centre was implemented.

Early project PPIE activity includes workshops at the command centre and control site to engage PPIE representatives to give lay perspectives on care coordination in hospitals, to inform the development of interview questions for hospital staff, and to suggest measures of patient safety. Representatives have been recruited by advertisement through the patient groups both associated and not associated with the Bradford hospital site. Representatives will be reimbursed in accordance with the NIHR standards and the INVOLVE framework, for example, monetary or voucher reimbursement for contribution to workshops and additional reimbursement remote participation. Our PPIE co-applicant will support qualitative data analysis through review and further development of emerging themes in the dataset. Towards the end of the project, a joint workshop will host PPIE representatives from both hospital sites to help interpret findings and to draft a PPIE communication plan. Our PPIE Lay Leader and project co-applicant will also co-develop case descriptions with the qualitative research team and will coauthor all publications to provide a PPIE perspective. We will provide ongoing feedback as part of our engagement

work with local trust stakeholders, including patient representatives

## Ethics and dissemination

This protocol has been approved by the University of Leeds Engineering and Physical Sciences Research Ethics Committee (#MEEC 20-016) and the NHS Health Research Authority (IRAS No.: 285933). The protocol was developed by a multidisciplinary team of coapplicants, including PPIE representatives, and by the NIHR as part of the NIHR 19/16 HSDR Digital Technologies to Improve Health and Care funding processes.

At the end of the study, we will provide a report detailing design feedback to improve the implementation of the AI command centre at the Bradford site and more generally for the system supplier and the digital technology industry. Our results will be communicated through peer-reviewed publications in international journals and conferences.

To contribute to objective 4 ('synthesis and dissemination'), we will involve the research team, project co-applicants, the study steering committee, stakeholders and PPIE representatives to consolidate research findings, PPIE perspectives and study steering committee insight into outputs to appropriate audiences.

After the project has completed, data that is approved to do so will be offered to the University of Leeds Research Data Repository, in accordance with our Data Management Plan (online supplemental appendix 3).

## Limitations

This proposed protocol was planned prior to the COVID-19 pandemic, which has caused substantial changes to the structures and processes used in healthcare systems. Our substudies on data quality, patient flow and patient safety are intended to provide quantification of the influence of the command centre implementation but will be limited in their capability to distinguish such contributions from those motivated by the response to COVID-19. Our mixed-method approach and involvement from our international study steering group will help to define the context of this turbulent period and to describe the processes of change in the hospitals studied. Under the epistemic constraints of our pre-COVID, funder-approved protocol, we will interpret our research through these contextual descriptions.

**Author affiliations**
¹School of Computing, University of Leeds Faculty of Engineering and Physical Sciences, Leeds, UK
²Bradford Royal Infirmary, Wolfson Centre for Applied Health Research, Bradford, UK
³School of Psychology, University of Leeds Faculty of Social Sciences, Leeds, UK
⁴Department of Computer Science, University of York, York, UK
⁵Bradford Royal Infirmary, Bradford Teaching Hospitals NHS Foundation Trust, Bradford, UK
⁶Faculty of Health Studies, University of Bradford, Bradford, UK

**Acknowledgements** We thank the PPIE representatives for their contribution to the conception, development, ongoing implementation and future communication of this project. This research is supported by the National Institute for Health Research (NIHR) Yorkshire and Humber Patient Safety Translational Research Centre (NIHR Yorkshire and Humber PSTRC).

**Contributors** JB, CiM, CaM and OJ conceptualised the study. NS contributed a patient and public perspective for decisions about research design, acceptability, relevance, conduct and governance from study conception to dissemination plans. JB, CaM, CiM, NS, IH, TL, RR and OJ contributed to the design of the protocol, reviewed the initial funding bid and responded to funder queries. CiM and CaM prepared the first draft of the protocol manuscript and TFM led revisions. All authors involved in revising the work for important intellectual content and approved the final versions for publication.

**Funding** This project is funded by the National Institute for Health Research Health Service and Delivery Research Programme (NIHR129483).

**Disclaimer** The views expressed in this article are those of the authors and not necessarily those of the NHS, the NIHR, or the Department of Health and Social Care.

**Competing interests** None declared.

**Patient consent for publication** Not applicable.

**Provenance and peer review** Not commissioned; externally peer reviewed.

**ORCID iDs**
Ciarán McInerney http://orcid.org/0000-0001-7620-7110
Teumzghi F Mebrahtu http://orcid.org/0000-0003-4821-2304
Rebecca Randell http://orcid.org/0000-0002-5856-4912

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
