## [Reviewer comments · BMJ Open]

ARTICLE DETAILS

TITLE (PROVISIONAL)	Evaluating the safety and patient impacts of an Artificial Intelligence Command Centre in acute hospital care: A mixed-methods protocol
AUTHORS	McInerney, Ciarán; McCrorie, Carolyn; Benn, Jonathan; Habli, Ibrahim; Lawton, Tom; Mebrahtu, Teumzghi; Randell, Rebecca; Sheikh, Naeem; Johnson, Owen

VERSION 1 – REVIEW

REVIEWER	Arora, Anmol University of Cambridge, School of Clinical Medicine
REVIEW RETURNED	11-Aug-2021

GENERAL COMMENTS	This is interesting and important study. The protocol is well-written and well-considered. Minor suggestions on what can be expanded upon in the manuscript include: 1. More specific detail about the AI that will be used in the Artificial Intelligence Command Centre and what specific variables might be included.2. More specific detail about PPIE including recruitment of representatives, reimbursement and adherence to NIHR PPIE standards (https://www.nihr.ac.uk/documents/payment-guidance-for-members-of-the-public-considering-involvement-in-research/27372, https://sites.google.com/nihr.ac.uk/pi-standards/home).
---

REVIEWER	Sumner, Jennifer Alexandra Hospital, Medical Affairs - Research, Innovation & Enterprise
REVIEW RETURNED	17-Aug-2021

GENERAL COMMENTS	Thank you for the opportunity to review the proposed study of an acute hospital command centre. The protocol is comprehensive and well written. I enclose only a few minor comments for consideration. 1. It would be useful to the reader to include some more background on the Bradford AI command centre to give a little more context for the evaluation. For example, when was the system introduced at the hospital, how large is the hospital it serves, what data does the AI system use and what are the outputs, is the system only applicable to certain departments e.g., acute admissions...Please include what you think is relevant
--

	2. Sub study 1: You report that the qualitative sub studies will inform the identification of variables that you will assess for data quality. This is quite broad, could you specify what domains you will look at e.g., clinical data? Operational metrics etc. etc. For clarity could you also specify which specific qualitative activities will inform variable selection 3. You plan to draw comparisons with a site from the same geographically location – is there potential for contamination bias? i.e., do staff rotate/serve at multiple institutions? 4. You plan to make observations in the command centre. Will these observations be limited to the physical ‘command centre’ or extend to on the ground observations i.e., how use of the command centre influences care practice?
--	--

VERSION 1 – AUTHOR RESPONSE

Reviewer: 1

Mr. Anmol Arora, University of Cambridge

Comments to the Author:

This is interesting and important study. The protocol is well-written and well-considered. Minor suggestions on what can be expanded upon in the manuscript include:

COMMENT 1: More specific detail about the AI that will be used in the Artificial Intelligence Command Centre and what specific variables might be included.

AUTHORS’ RESPONSE: Thank you. We have added description of the AI to be used in the command centre (see paragraph six of the introduction section). We have also included a table with the list of variables we intend to include (see Table 1 in the methods section)

COMMENT 2: More specific detail about PPIE including recruitment of representatives, reimbursement and adherence to NIHR PPIE standards (<https://www.nihr.ac.uk/documents/payment-guidance-for-members-of-the-public-considering-involvement-in-research/27372>, <https://sites.google.com/nihr.ac.uk/pi-standards/home>).

AUTHORS’ RESPONSE: Thank you for the feedback. Yes, we have adhered to the NIHR guidance and INVOLVE framework. We have added this detail in the newly-highlight section on PPIE.

Reviewer: 2

Dr. Jennifer Sumner, Alexandra Hospital, NUS Yong Loo Lin School of Medicine

Comments to the Author:

Thank you for the opportunity to review the proposed study of an acute hospital command centre. The protocol is comprehensive and well written. I enclose only a few minor comments for consideration.

COMMENT 1: It would be useful to the reader to include some more background on the Bradford AI command centre to give a little more context for the evaluation. For example, when was the system introduced at the hospital, how large is the hospital it serves, what data does the AI system use and what are the outputs, is the system only applicable to certain departments e.g., acute admissions...Please include what you think is relevant

AUTHORS’ RESPONSE: We thank the reviewer for their positive comments on the presentation of the manuscript. We have now included details about the command centre and the hospital it serves—see paragraph six of the introduction section.

COMMENT 2: Sub study 1: You report that the qualitative sub studies will inform the identification of variables that you will assess for data quality. This is quite broad, could you specify what domains you will look at e.g., clinical data? Operational metrics etc. etc. For clarity could you also specify which specific qualitative activities will inform variable selection?

AUTHORS' RESPONSE: Thank you. We have clarified now that the qualitative sub-study 4 will help to define the tiles of interest which are largely operational metrics. The final included set of variables will be defined based on availability of historic data, data quality and relationship with the intervention logic, as established through the parallel qualitative work.

COMMENT 3: You plan to draw comparisons with a site from the same geographically location – is there potential for contamination bias? i.e., do staff rotate/serve at multiple institutions?

AUTHORS' RESPONSE: Thank you for raising this important issue. The intervention and control sites are two different hospitals with no staff rotation/serving on both institutions. Hence, we believe that any contamination is very minimal. However, the qualitative data collection at the control site will explore any potential links to the intervention site.

COMMENT 4: You plan to make observations in the command centre. Will these observations be limited to the physical 'command centre' or extend to on the ground observations i.e., how use of the command centre influences care practice?

AUTHORS' RESPONSE: Thank you for raising the important issue of observing the impact of the command centre beyond the centre itself. This was indeed a feature of our proposal and we will undertake observations in both physical command centre and elsewhere in the hospital to understand how the command centre influences care practice (see sub-study 4 “structured observation”)

VERSION 2 – REVIEW

REVIEWER	Arora, Anmol University of Cambridge, School of Clinical Medicine
REVIEW RETURNED	15-Nov-2021

GENERAL COMMENTS	I am happy that the authors have adequately addressed my comments from the first round of peer review and congratulate them on a well-written and important manuscript.
---

REVIEWER	Sumner, Jennifer Alexandra Hospital, Medical Affairs - Research, Innovation & Enterprise
REVIEW RETURNED	03-Dec-2021

GENERAL COMMENTS	The revisions address all comments. All the best with the project.
---